

# Population structure and genetic diversity of *Tamarix chinensis* as revealed with microsatellite markers in two estuarine flats

Zhao-Yu Jiang[*], Ao-Ao Yang[*], Hai-Guang Zhang, Wen-Bo Wang and Ru-Hua Zhang

Linyi University, College of Life Science, Linyi City, Shandong Province, China
[*] These authors contributed equally to this work.

## ABSTRACT

**Background**. *Tamarix chinensis* Lour. is a 3–6-meter-tall small tree with high salt- and alkali- tolerance and aggressive invasiveness, mainly distributed in the eastern part of China in warm-temperate and subtropical climate zones, yet there is little information available regarding genetic diversity and population structure.

**Methods**. A total of 204 individuals of nine *T. chinensis* populations were investigated for genetic diversity and population structure using a set of 12 highly polymorphic microsatellite markers.

**Results**. The total number of alleles detected was 162, the average number of effective allele was 4.607, the average polymorphism information content (PIC) value of the 12 loci was 0.685, and the mean observed heterozygosity (Ho) and the mean expected heterozygosity (He) was 0.653 and 0.711, respectively. Analysis of molecular variance (AMOVA) showed a 5.32% genetic variation among *T. chinensis* populations. Despite a low population differentiation, Bayesian clustering analysis, discriminant analysis of principal components (DAPC) and the unweighted pair group method with arithmetic mean (UPGMA) clearly identified three genetic clusters correlated to the populations' geographic origin: the northern populations including those from Yellow River Delta, the Fangshan (FS) population from Beijing, the Changyi (CY) population from Bohai Bay, the Huanjiabu (HHJ) population from Hangzhou Bay, and the remaining two populations from Hangzhou Bay. There was a significant relationship between the genetic distance and geographical distance of the paired populations. Gene flow (Nm) was 4.254 estimated from $F_{ST}$.

**Conclusion**. *T. chinensis* possessed high genetic diversity comparable to tree species, and although the population differentiation is shallow, our results classified the sampled populations according to sampling localities, suggesting the different origins of the study populations.

Corresponding author
Ru-Hua Zhang,
ruhuazh2000@126.com

## INTRODUCTION

The old world temperate genus *Tamarix* L. (Tamaricaceae) comprises 68 species native to Eurasia and Africa, typically inhabiting deserts and sub-deserts in temperate and subtropical regions (*Baum, 1978*; *Zohary, 1987*). Originating at least prior to Eocene in Mediterranean basin or Irano-Turanian regions where comparatively high number of species are found, the genus spread eastward and westward from the diversity centers (*Villar, Juan & Alonso, 2014*; *Qaiser, 1981*). *Tamarix* reached the western China in Oligocene (*Wang & Yang, 1980*; *Song, 1958*) and Tarim Basin is the secondary distribution center with 16 species (five being endemic) (*Zhang, Pan & Yin, 2003*; *Wang, 1992*). *Tamarix chinensis* and its two congeners (*T. austromongolica* and *T. ramosissima*) further colonized eastward, possibly along the Yellow River corridor according to molecular evidence, and it established populations along East coast of China (*Liang et al., 2018b*).

*T. chinensis* Lour. is a salt- and alkali- tolerant deciduous shrub with 3–6-meter-height with significant ecological and economic importance (*Jiang, Chen & Bao, 2012*; *Wang et al., 2009*). Through secreting excessive salts with glands in leaves, it can tolerate high salt concentration in soil and has been widely used for establishing coastal protection forests in the East Sea coast of China (*Zhao et al., 2008*). Its bark and twig has high tamarixone and tamarixolare content which can cure cold (*Jiang et al., 2011*). It is an important firewood source where coal is unaffordable, especially in some regions of western China. Its wood is hard and flexible and wear-resistant, which is often for weaving use. Its wood fiber can be used as raw material for paper and fiberboard. It blooms three times a year from May to October and is often used as a garden plant to beautify the environment. However, this species has rapidly dwindled in the last 20 years due to overexploitation for heating, weaving and medical use. Moreover, the taxonomic position of *T. chinensis* and *T. austromongolica* remains unsolved. Although *T. austromongolica* is distributed in the upper reaches of the Yellow River and *T. chinensis* in the lower reaches (*Liang et al., 2018b*), their phenotypic traits are quite close and some botanists placed them as the same species.

The genetic diversity of a species plays a crucial role in providing evolution potential and affecting its adaptation to various environmental selective pressures including temperature extremes, drought and flood, and high salt and alkali concentration (*Hughes et al., 2008*). Therefore, evaluating the genetic diversity within and among the sampled populations of a species is elementary for further clarifying the distribution of genetic variation across its range, estimating the gene flow levels between populations, and determining the evolutionary significant units (ESU) (*Jiang et al., 2019*; *Islam et al., 2012*). Up to data, several studies have been reported on the population diversity and structure of *T. chinensis* and its congeners. With six polymorphic microsatellite markers, *Liang et al. (2018a)* performed the genetic analysis of the populations collected along the Yellow River and revealed moderate levels of genetic diversity (a range of the mean expected heterozygosity (He) from 0.366 to 0.740 for the studied population) and low population genetic differentiation ($F_{ST} = 0.053$). Random primers analysis of populations distributed in the Yellow River Delta with Random Amplified Polymorphic DNA (RAPD) (*Zhao et al., 2008*) and Inter-simple Sequence Repeat (ISSR) (*Jiang, Chen & Bao, 2012*) also showed

low population genetic differentiation levels (Gst =0.0507 for RAPD analysis and Gst =0.0707 for ISSR) but comparatively high genetic diversity levels (Nei index =0.4061 for RAPD analysis and Nei index =0.276 for ISSR analysis). *Zhu et al. (2016)* conducted a correlation analysis between population genetic diversity and soil salinity, and the results showed that overall genetic diversity within populations decreased progressively along with the increasing soil salinity. Of note, *Lee et al. (2018a)* studied the population genetic structure and genetic diversity with SNPs derived from genotyping-by-sequencing (GBS) and identified genetic discontinuities among natural populations occurring in different river basins of the northwest of USA. Of studies on Chinese *T. chinensis* populations, no one included any of the extensively distributed populations in Hangzhou Bay, an estuarine flat in the subtropical zone.

Simple sequence repeat (SSR; microsatellite) markers are widely used in population diversity and structure of various trees and shrubs for their codominant characteristics, high polymorphism and stableness (*Fernádez et al., 2018*; *Nybom, 2004*). In this study, we used 12 SSR primers to detect and estimate the variation of *T. chinensis* mainly from two estuarine flats of different climate zones: Yellow River Delta of the warm-temperate zone and Hangzhou Bay of the subtropical zone. Our specific aims were to (1) evaluate the genetic diversity and genetic differentiation among nine *T. chinensis* populations, (2) determine whether population structure relative to sampling locations exists, and (3) compare the population genetic indices of the two different climate regions. Our study can deepen understanding of the spatial distribution pattern of genetic variation of *T. chinensis* and lay foundation for further exploring its adaptation to heterogeneous environments.

## MATERIALS AND METHODS

### Sample collection and DNA extraction

Samples were taken from nine natural populations of *T. chinensis* distributed from Beijing, Shandong Province and Zhejiang Province. Sampling mainly focused on two estuarine flats of different climate zones: Yellow River Delta of the warm-temperate zone, and Hangzhou Bay of the subtropical zone (Fig. 1, Table 1). These two estuarine flats have many *T. chinensis* populations of different sizes and ages and populations of Yellow River Delta usually have larger size than those of Hangzhou Bay. Four populations (YHK, YDG, YXX and YHD) from Yellow River Delta and three populations (HHJ, HLS and HCX) from Hangzhou Bay were sampled. One population (CY) of the Bohai Sea coast from Changyi City and one inland population (FS) from Beijing were also included. A total of 19–29 individuals per population were collected according to the population size and a total of 204 individuals from nine populations was sampled. We collected young leaves from individual trees spaced at least 50 m apart, in order to avoid sampling related individuals. Fresh leaves were desiccated insilica gel. Geographical coordinates and altitude were acquired with GPS (Table 1). A modified cetyltrimethyl ammonium bromide (CTAB) method (*Su et al., 1998*) was used to extract total genomic DNA. Quality and quantity of the extracted DNA were measured with 1.0% (w/v) agarose gel electrophoresis and a NanoDrop 2000c spectrophotometer (Thermo Fisher Scientific, Waltham, MA, USA), respectively. DNA was diluted to 20 ng/μl and stored at −20 °C for later use.

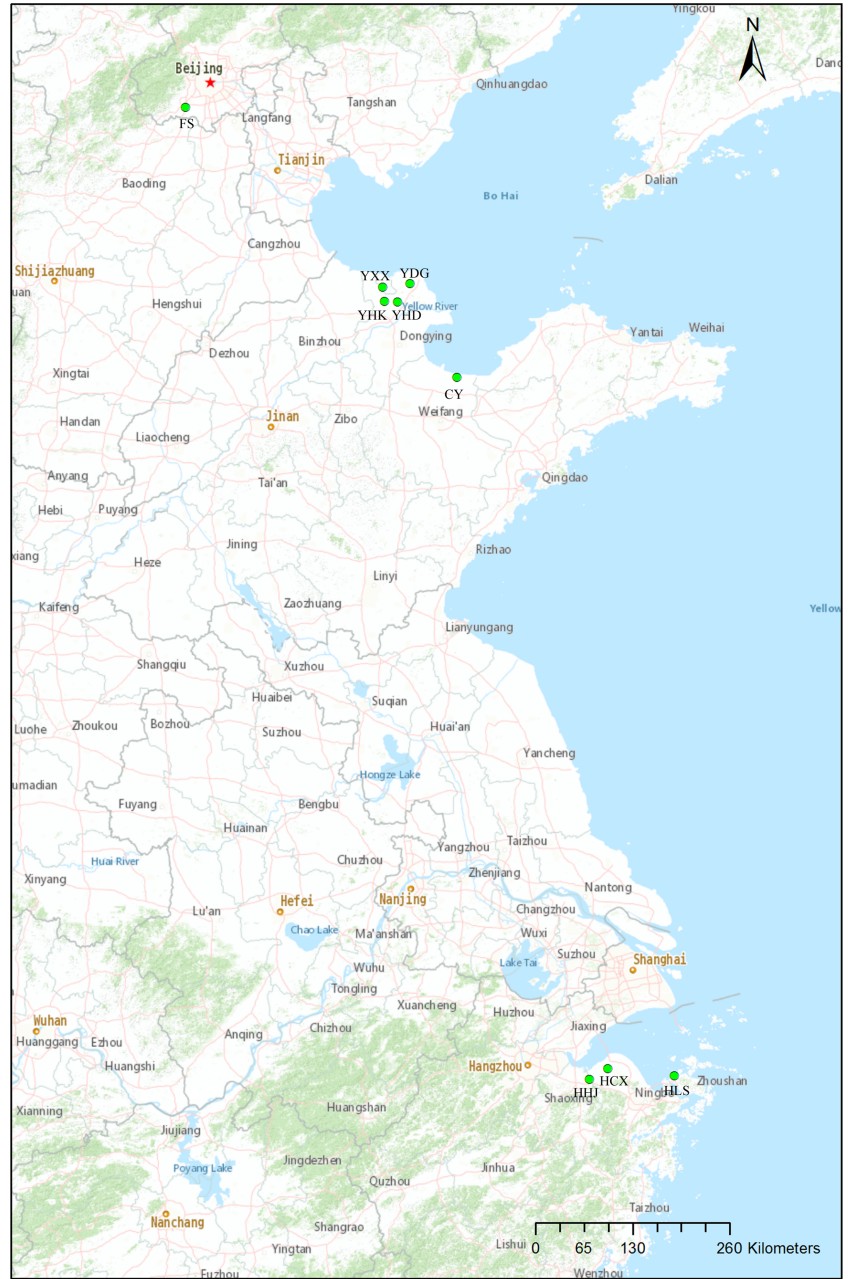

**Figure 1** Locations of the nine natural *Tamarix chinensis* populations. The map was produced with the software ArcGIS 10.5.

We used a set of 12 highly polymorphic SSR markers (Table 2) for population genetic analysis, six of which were EST-SSRs (*Zhang et al., 2011*) and the remaining six of which were genomic SSRs (*Zhang, Wen & Xu, 2019*). The polymerase chain reaction (PCR) was carried out according to (*Zhang, Wen & Xu, 2019*). Forward primers were labeled with four color fluorescent dyes (6-FAM, HEX, TAMRA and ROX) (Invitrogen, Shanghai, China).The PCR conditions of both kinds of SSR were as follows: 4 min denaturation at

**Table 1** Geographical information of nine natural *T. chinensis* populations in China.

| Populations | ID | Sample size | X (°N) | Y (°E) | Elevation |
|---|---|---|---|---|---|
| Helou Shandong | YHK | 23 | 37.853 | 118.491 | 2.1–2.6 |
| Changyi Shandong | CY | 24 | 37.127 | 119.356 | 1.9–2.3 |
| Fangshan Beijing | FS | 20 | 39.674 | 116.095 | 48.3–47.7 |
| Dongyinggang Shandong | YDG | 24 | 38.02 | 118.795 | 1.8–2.9 |
| Xinhu Shandong | YXX | 22 | 37.988 | 118.467 | 2.7–3.3 |
| Hekoudong Shandong | YHD | 22 | 37.847 | 118.542 | 2.0–3.2 |
| Huangjiabu Zhejiang | HHJ | 22 | 30.222 | 120.952 | 1.1–2.2 |
| Linshan Zhejiang | HLS | 19 | 30.239 | 121.972 | 1.7–2.3 |
| Cixi Zhejiang | HCX | 29 | 30.445 | 121.172 | 1.4–2.2 |

**Table 2** Genetic diversity among 12 simple sequence repeat (SSR) loci of nine *T. chinensis* populations.

| Locus | Na | Ne | Ho | He | I | PIC | $F_{ST}$ | Nm | Source |
|---|---|---|---|---|---|---|---|---|---|
| CF199044 | 10 | 5.948 | 0.8162 | 0.8341 | 1.9136 | 0.810 | 0.0437 | 5.471 | *Zhang et al. (2011)* |
| EH051607 | 13 | 4.014 | 0.623 | 0.7529 | 1.6869 | 0.716 | 0.0953 | 2.373 | *Zhang et al. (2011)* |
| EH053495 | 18 | 3.041 | 0.755 | 0.6728 | 1.4644 | 0.628 | 0.0421 | 5.688 | *Zhang et al. (2011)* |
| CV790971 | 9 | 2.848 | 0.5373 | 0.6505 | 1.3133 | 0.592 | 0.0539 | 4.388 | *Zhang et al. (2011)* |
| EG967460 | 23 | 7.428 | 0.7407 | 0.8677 | 2.3415 | 0.854 | 0.0417 | 5.745 | *Zhang et al. (2011)* |
| AB298390 | 33 | 10.637 | 0.8469 | 0.9083 | 2.7643 | 0.899 | 0.0684 | 3.405 | *Zhang et al. (2011)* |
| MG856344 | 15 | 4.618 | 0.7487 | 0.7855 | 1.8329 | 0.758 | 0.0491 | 4.842 | *Zhang, Wen & Xu (2019)* |
| MG856346 | 13 | 5.545 | 0.6546 | 0.8218 | 1.9932 | 0.787 | 0.0582 | 4.046 | *Zhang, Wen & Xu (2019)* |
| MG856349 | 15 | 4.089 | 0.7473 | 0.7731 | 1.4882 | 0.745 | 0.0728 | 3.814 | *Zhang, Wen & Xu (2019)* |
| Gssr3 | 4 | 2.478 | 0.501 | 0.5361 | 0.7687 | 0.477 | 0.110 | 2.023 | unpublished |
| Gssr4 | 3 | 2.312 | 0.3939 | 0.4073 | 0.5401 | 0.406 | 0.075 | 3.083 | unpublished |
| Gssr5 | 6 | 2.322 | 0.4752 | 0.5230 | 0.7210 | 0.553 | 0.036 | 6.694 | unpublished |
| mean | 13.50 | 4.607 | 0.6533 | 0.7111 | 1.569 | 0.685 | 0.0622 | 4.245 | |

**Notes.**

Na, number of alleles; Ne, effect number of alleles; Ho, observed heterozygosity; He, expected heterozygosity; I, Shannon's information index; PIC, polymorphism information content; Fst, genetic differentiation coefficient; Nm, gene flow.

94 °C; followed by 30 cycles of 94 °C denaturation for 30s, varied annealing temperatures ranging from 49.6–58.3 ° C for 30s (Table S1), and an elongation at 72 °C for 30s; and a final extension step at 72 °C for 5 min. PCR products were run on an ABI3730 XL Analyzer (Applied Biosystems, Foster City, CA, USA) and allele sizes were determined with GeneMapper 4.0 software (Applied Biosystems).

## Data analysis

The occurrence of null alleles has a biased effect on the estimation of population genetic parameters, so we used MICROCHECKER v2.2.3 (*Van Oosterhout et al., 2004*) to check for the presence of null alleles for all loci. Hardy-Weinberg equilibrium exact tests and multilocus linkage disequilibrium tests were performed with GENEPOP (version 1.2) (*Raymond & Rousset, 1995*), and sequential Bonferroni correction tests were applied to adjust the significance levels of genotypic linkage disequilibrium. GenAlEx 6.502
(*Peakall & Smouse, 2005*) was used to calculate genetic diversity indices including observed heterozygosity (Ho), expected heterozygosity (He), number of allele (Na), number of effective alleles (Ne), inbreeding coefficient ($F_{IS}$), and Shannon diversity index (I). The genetic differentiation coefficient (Fst) across all populations at each locus and over all loci, gene flow (Nm = $(1 − F_{ST})/4\,F_{ST}$), and allelic richness (Ar) were calculated using FSTAT version2.9.3 software (*Goudet, 2001*). We used CERVUS version 3.0.3 (*Kalinowski, Taper & Marshall, 2007*) to calculate the polymorphism information content (PIC) value of each locus according to: $PIC = 1 − \sum_{i=1}^{n} f_i^2$ where $f_i$ is the frequency of the *i*th allele and *n* is the allele number.

Analysis of molecular variance (AMOVA) was performed using ARLEQUIN 3.5 (*Excoffier & Lischer, 2010*) to partition the genetic variability into within- and among-population components. The significance was tested based on 1,000 permutations for the AMOVA results (*Excoffier, Laval & Schneider, 2005*). Bottleneck events of the studied populations were tested using BOTTLENECK 1.2.02 (*Piry, Luikart & Cornuet, 1999*) with stepwise mutation model (SMM) and two-phased mutation model (TPM). Recently bottlenecked populations have higher observed heterozygosity (unbiased gene diversity) than expected from the observed number of alleles across microsatellite loci. Variance for TPM was set to 30 and the proportion of SMM in TPM was set to 80%. The "mode-shift" of allele frequency distribution was also used to identify bottlenecked populations (*Luikart et al., 1998*).

Bayesian clustering analysis was performed using STRUCTURE v2.3.4 software (*Pritchard, Stephens & Donnelly, 2000*) to determine the number of genetically homogeneous groups of the collected individuals and to evaluate the amount of admixture between individuals with the admixture model and allele frequencies correlated. We run the program with 100,000 burn-in steps followed by 100,000 Markov chain Monte Carlo (MCMC) steps, and 10 independent runs. The optimal K was determined through the online program STRUCTURE HARVESTER (*Earl & Von Holdt, 2012*) for each K from 1 to 10. Graphics were displayed with the DISTRUCT program (*Rosenberg, 2004*).We also implemented the Discriminant Analysis of Principal Components (DAPC) (*Jombart, Devillard & Balloux, 2010*) using the "*adegenet*" package (*Jombart, 2008*) for software R version 3.4.2 (*R Core Team, 2017*) to evaluate population structure. DAPC is a multivariate approach that is free of assumptions of Hardy–Weinberg equilibrium (HWE) and maximizes the genetic differentiation between groups with unknown prior clusters, thus improving the discrimination of populations. A dendrogram was also constructed using the unweighted pair-group method with arithmetic means (UPGMA) algorithm based on Nei's genetic distance among populations with the NTSYS version 2.1 software (*Rohlf, 2000*).

We used the mantel function in the "*vegan*" R package (*Dixon, 2003*) to test the correlation between genetic distance (pairwise $F_{ST}$) and Euclidean geographical or environmental distance. The climatic layers of 19 bioclimatic variables under current periods were downloaded from the WorldClim database (http://www.worldclim.org/) with a resolution of 2.5 arc-minutes to evaluate the effects of environment on genetic variation. The Vifstep function of the R package usdm (*Naimi et al., 2014*) was used to select the

geographical and environmental variables with variance inflation factor (VIF) below 10 for Euclidean distance calculation. VIF measures the collinearity of the variables and the larger the VIF, the stronger the linear relationship between two or more predictor variables. After forward selection, five bioclimate variables (Bio2, Bio5, Bio8, Bio15 and Bio18) were chosen to constitute the environmental variable matrix. Euclidean distances of the five chosen environmental variables from different populations constituted the environmental variable matrix for the Mantel test. The significance of the correlation tests was evaluated with 9,999 permutations. To determine whether geographical or environmental variables may have affected genetic distance, we also conducted a partial Mantel test using the mantel.partial function in the R package vegan with 9,999 permutations.

# RESULTS

## Genetic diversity

The micro-checker analysis revealed one locus in population FS and population HHJ, three loci in population HCX, four loci in population YHD, five loci in population YHK with null alleles, while no null alleles were detected in four populations (CY, YDG, YXX, and HLS) (Table S2). Specifically, loci 2, 4, 5, 9, 11 revealed null alleles in two populations; loci 3, 8 and 10 in one population, while loci 1, 6, 7, 12 showed no null alleles. Hardy-Weinberg exact tests showed two loci in one populations (YXX), three loci in four populations (CY, FS, YHD, and HCX), four loci in two populations (YDG and HHJ), five loci in one population (YHK), and seven loci in one population (HLS) that deviated from equilibrium at $p < 0.05$ or at $p < 0.01$ (Table S3A). At population level, only one population (CY) was in HWE, and other populations deviated from equilibrium at $p < 0.05$ or $p < 0.01$ (Table S3B). Except for populations YDG, YXX and YHD, genotypic linkage disequilibrium (LD) was detected with a range of one locus pair (for population HYD) to 16 locus pairs (for population HHJ) after Bonferroni correction (Table S4A). Across all populations, 12 out of 132 locus pairs showed linkage disequilibrium, with loci 2 and 4 having the lowest number of pairs (1 pair) in LD (Table S4B).

We used 12 pairs of SSR primers to generate 162 alleles from 204 individuals of nine *T. chinensis* populations (Table 2). The results showed an average value of 13.50 for all alleles ranging from three (Gssr4) to 33 (AB298390) per locus. The effective number of alleles (*Ne*) ranged from 2.312 (Gssr4) to 10.636 (AB298390), with an average of 4.607. The observed heterozygosity (*Ho*) for each locus varied from 0.3939 (Gssr4) to 0.8469, with an average of 0.6533 and the expected heterozygosity (*He*) varied from 0.4073 (Gssr4) to 0.9083 (AB298390), with an average of 0.7111. Shannon's information index (*I*) ranged from 0.5401 (Gssr4) to 2.7643 (AB298390), with an average of 1.569. The genetic differentiation coefficient (Fst) (0.036 (Gssr5)–0.110 (Gssr3)) and the polymorphic information content (*PIC*) values (0.406 (Gssr4) −0.899 (AB298390)) were detected at all loci. Gene flow (Nm) ranged from 2.023 (Gssr3) to 6.694 (Gssr5), with an average of 4.245.

The genetic diversity analysis of nine *T. chinensis* population of was shown in Table 3. Allele richness ranged from 8.56 (HHJ) to 10.38 (YXX) with an average of 9.37. The number of alleles (Na) varied from 7.25 for the HHJ population to 9.5 for the YXX population

**Table 3  Genetic diversity parameters at the population level based on 12 SSR markers.**

| Population | Na | Ne | Ar | Np | I | Ho | He | Fis |
|---|---|---|---|---|---|---|---|---|
| YHK | 8.000 | 4.984 | 9.25 | 2 | 1.651 | 0.570 | 0.690 | 0.177 |
| CY | 7.375 | 4.516 | 9.53 | 2 | 1.585 | 0.668 | 0.678 | 0.022 |
| FS | 7.375 | 4.022 | 8.85 | 3 | 1.529 | 0.667 | 0.655 | −0.021 |
| YDG | 8.375 | 5.325 | 9.84 | 3 | 1.727 | 0.746 | 0.712 | −0.072 |
| YXX | 9.500 | 5.914 | 10.38 | 7 | 1.854 | 0.743 | 0.739 | −0.011 |
| YHD | 8.375 | 4.953 | 9.85 | 7 | 1.693 | 0.687 | 0.700 | 0.018 |
| HHJ | 7.250 | 3.978 | 8.56 | 5 | 1.547 | 0.749 | 0.678 | −0.099 |
| HLS | 7.750 | 4.560 | 8.76 | 6 | 1.648 | 0.586 | 0.702 | 0.148 |
| HCX | 8.375 | 5.047 | 9.27 | 16 | 1.698 | 0.662 | 0.706 | 0.061 |
| mean | 8.042 | 4.811 | 9.37 | 5.667 | 1.659 | 0.674 | 0.696 | 0.025 |

**Notes.**

Abbreviations: Na, the number of different alleles; Ne, the number of effective alleles; Ar, the allelic richness; Np, the number of private alleles; I, Shannon's information index; Ho, the observed heterozygosity; He, the expected heterozygosity; Fis, inbreeding coefficient.

with an average of 8.042 and the effective number of alleles (Ne) varied from 3.978 for the HHJ population to 5.915 for the YXX population with an average of 4.811. Shannon's information index (I) varied from 1.547 (HHJ) to 1.854 (YXX), with a mean of 1.659. Mean value of expected heterozygosity (He = 0.696) was slightly higher than that of observed heterozygosity (Ho =0.674). The fixation index (Fis) ranged from −0.099 for the HHJ population to 0.177 for the HHK population, with an average of 0.025. Out of the nine studied populations, population YXX from Yellow River Delta showed the highest values of genetic diversity parameters except for the number of private allele (Np =7), which was only lower than that of population HCX (Np =16). Altogether, 54 private alleles were identified and populations YHK and CY had the lowest number (Np =2) while population HCX displayed most private alleles (Np =16).

Averagely, higher values of genetic diversity indices (Na =8.5625, Ne =5.294, $I = 1.73125$, Ho =0.680, He =0.710) for populations from Yellow River Delta were obtained than those of populations from Hang Zhou Bay (Na =7.7916, Ne =4.528, $I = 1.631$, Ho =0.666, He =0.695) except for the number of private allele (Np =4.75 for populations from Yellow River Delta and Np =9 for populations from Hangzhou Bay) (Table S8). Inbreeding coefficients of populations from Yellow River Delta (−0.072−0.017, average =0.028) are lower than those of populations from Hangzhou Bay (−0.099−0.148, average = 0.0367).

Under SMM and TPM, our Wilcoxon tests detected no significant heterozygote excess for the nine populations (Table S7). Likewise, mode-shift tests revealed an L-shaped distribution of alleles, suggesting the absence of a recent bottleneck.

## Population differentiation and genetic structure

Low population genetic differentiation ( $F_{ST} = 0.0532$, $p < 0.001$) (Table 4) was detected across *T. chinensis* populations. $F_{ST}$ values of the 12 loci ranged from 0.036(for Gssr5) to 0.110(for Gssr3) (Table 2). AMOVA test revealed significant differentiation detected among regions (Fct =0.0252, $p < 0.05$), among populations within regions (Fsc =0.028,

**Table 4** The partition of SSR variation of *T. chinensis* by analysis of molecular variance (AMOVA).

| Source of variation | d.f. | Sum of squares | Variation components | Percentage of variation |
|---|---|---|---|---|
| Among Regions | 3 | 43.127 | 0.08068 Va | (Fsc) 2.8[*] |
| Among Pops within regions | 5 | 35.648 | 0.08787 Vb | (Fct) 2.52[**] |
| Within populations | 403 | 1,242.834 | 3.08396 Vc | 94.68[**] |
| Total | 411 | 3.252 | 100% | |

Notes.
[*]$p < .05$.
[**]$p < .01$.

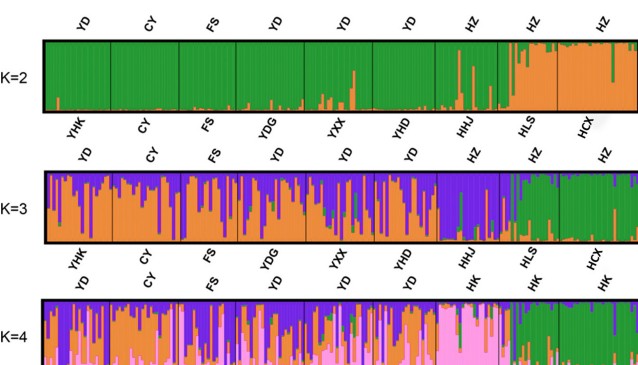

**Figure 2** Individual and population memberships to genetic clusters for $k = 2, 3$ and 4 using STRUCTURE.

$p < 0.001$), and 94.68% of variation was within-populations. The pairwise $F_{ST}$ values between populations are shown in Table S5 . The genetic differentiation between the CY and HCX populations was highest at 0.110.

Bayesian-based structure analysis supported the populations studied were classified into two clusters almost without admixture with a maximum delta value of 72.40 at $K = 2$ (Table S6). Populations from Yellow River Delta, CY population, FS population from Beijing, and one population from Hangzhou Bay (HHJ) formed the first cluster, and the two other populations, Hangzhou Bay HCX and HLS, formed the second cluster (Fig. 2A). At $K = 3$, HHJ population of the first cluster was assigned to the third cluster only with 2 admixed individuals (Fig. 2B), which is consistent with the DAPC results. At $K = 4$, HHJ population and two other Hangzhou Bay populations (HLS and HCX) formed two distinct clusters almost without admixture, while the individuals from the remaining populations were mostly admixed with two other clusters with approximate proportion (Fig. 2C).

The nine *T. chinensis* populations were grouped into two groups based on UPGMA (Fig. 3), two populations from Zhejiang Province, and the remaining populations. HHJ population was grouped into the second group with populations from Shandong Province and Beijing, but with the shortest branch length to populations of the first group. DAPC identified three distinct groups (Fig. 4), which is consistent with structure result at $K = 3$.

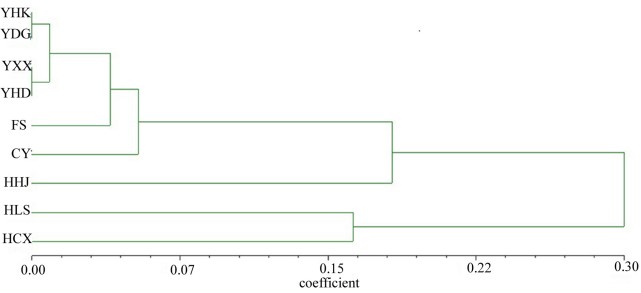

**Figure 3 UPGMA tree of the nine *T. chinensis* populations using Nei's genetic distance.**

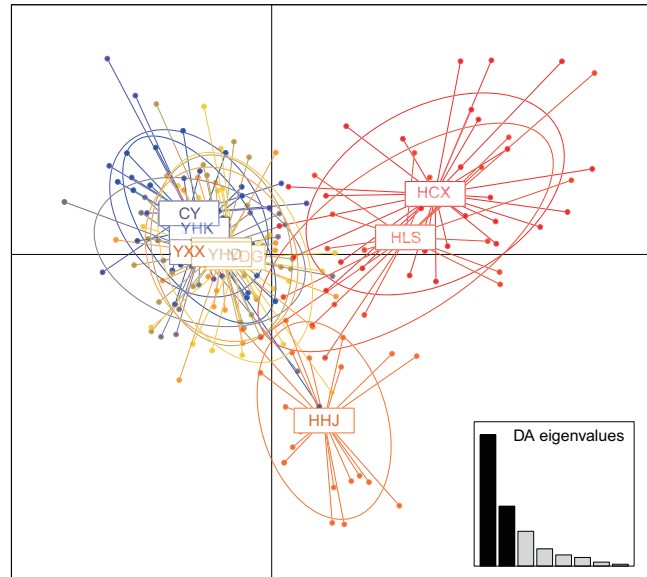

**Figure 4 Clustering results of *T. chinensis* populations obtained by discriminant analysis of principal components (DAPC, PCs = 40).**

## Mantel test

Mantel tests showed the studied populations conformed to the patterns of isolation by distance (IBD) ($R^2 = 0.5784$, $p = 0.012$) (Fig. 5A) and isolation by environment (IBE) ($R^2 = 0.5923$, $p = 0.015$) (Fig. 5B) at significant levels. Given the autocorrelation of geographical distance and environmental distance ($R^2 = 0.9357$, $p = 0.002$) (Fig. 5C), partial mantel tests were performed, which detected highly significant IBD ( $r = 0.7324$, $p = 0.002$), but no IBE patterns in the populations studied.

## DISCUSSION

In this study, we evaluated the genetic diversity level and population genetic structure of *T. chinensis* mainly distributed in two coastal flats of different climate zones. Our study revealed a high genetic diversity and low differentiation level and strong population genetic

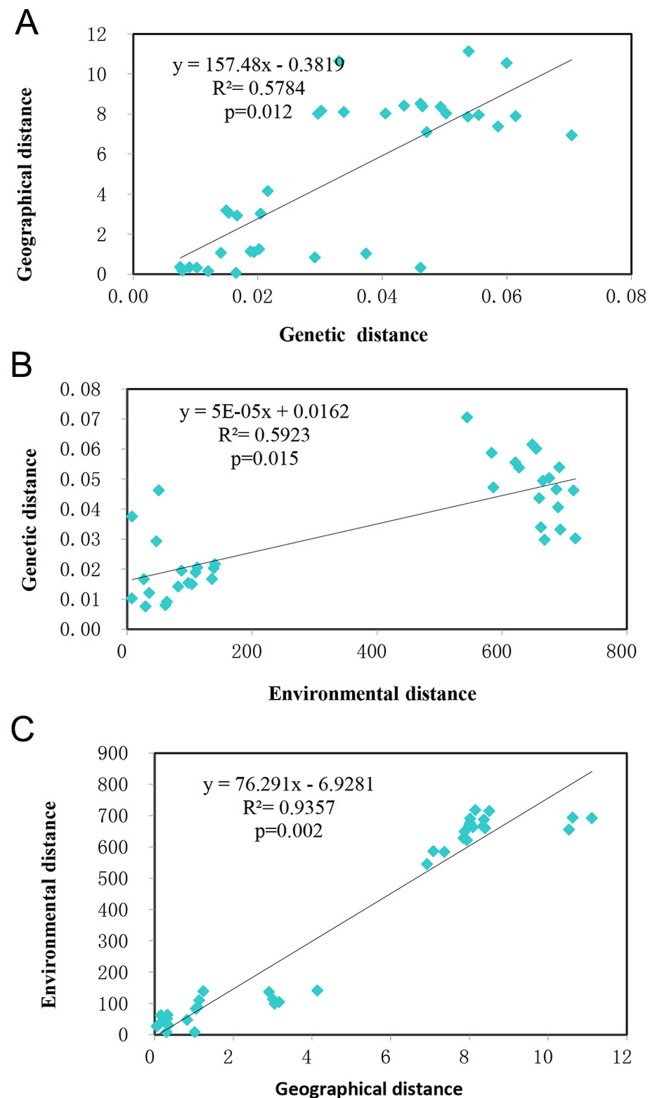

**Figure 5 The relationship of genetic, geographical, and environmental distances.** (A) The relationship between genetic distance and environmental distance of *T. chinensis*. (B) The relationship between genetic distance and environmental distance of *T. chinensis*. (C) The relationship between geographical distance and environmental distance of *T. chinensis*.

structure correlated to populations' location that previous studies did not detect. The genetic diversity level of populations from Yellow River Delta is higher than that of those from Hangzhou Bay on average

## High genetic diversity of *T. chinensis*

With highly polymorphic microsatellite markers, we detected high genetic diversity of *T. chinensis* (12 nSSRs, He =0.7111) (Table 2) comparable to the widely distributed long-lived tree species including *Quercus acutissima* (12 nSSRs, He = 0.760) (*Zhang et al., 2013*), *Q. liaotungensis* (19 nSSRs, He =0.801) (*Guo et al., 2021*), *Q. rubra* (15 SSRs, He =0.78) (*Lind*

*& Gailing, 2014*), and *Q. ellipsoidalis* (15 SSRs, He =0.76) (*Lind & Gailing, 2014*). Similarly, population genetic analysis through dominant markers including RAPD and ISSR also detected high genetic diversity in this species (Nei index =0.4061 for RAPD analysis and Nei index =0.276 for ISSR analysis) (*Jiang, Chen & Bao, 2012*; *Zhao et al., 2008*). The high level of genetic diversity of *T.chinensis* may be attributable to its high outcrossing rate, wide distribution, and long life span. Generally, the widely distributed long-lived plant species often maintain relatively high genetic diversity compared with narrow geographical ranges or endangered species (*Wang, Bernhardsson & Ingvarsson, 2020*; *Hamrick, Godt & Shermann-Broyles, 1992*). *T. chinensis* is an insect-pollinated plant with high outcrossing rate, although sometimes its pollen is dispersed *via* wind to the stigma of another plant. Our results showed there was no clonal reproduction in this species (data not shown) despite that it can be reproduced by cuttings and root suckers. It has a short generation time of 1-2y and its raceme flowers produce a large number tiny hairy seeds which can easily be transported by wind and/or water. Its seed can tolerate high salt concentration (0.15 g/l) (*Zhang & Zhang, 2019*) for a long time.

Our measurement of population genetic diversity showed significantly higher level than that of *Liang et al. (2018a)* where the genetic diversity ranged from 0.366 to 0.740 for populations along the Yellow River at six SSR markers and that of *Zhu et al. (2016)* where an average population heterozygosity of 0.493 was detected with 7 SSR markers in 26 populations from Yellow River Delta, and also that of its congener *T. taklamakanensis* (7SSRs, He = 0.432) (*Su et al., 2017*). The relatively low genetic diversity detected by those *T. chinensis* studies may be due to their usage of the microsatellite markers sets which developed through biotinylated-oligonucleotide capture method from *T. ramosissima* Ledeb. (*Gaskin, Pepper & Manhart, 2006*). These 10 microsatellite markers actually showed relatively low polymorphism, altogether two of which amplified length-variant alleles when PCR products were separated on PAGE gels (*Zhang, 2011*).

## Population genetic differentiation and structure

*T. chinensis* populations showed a low level of population differentiation ($F_{ST} = 0.0532$) (Table 2), in comparison with the result from its congener *T. taklamakanensis* (Fst =0.21) (*Su et al., 2017*). The level of population differentiation was consistent with findings from two population genetic studies of *T. chinensis* with SSR markers: (*Zhu et al., 2016*) ($F_{ST} = 0.0533$) and *Liang et al. (2018a)* ($F_{ST} = 0.058$). The low level of population differentiation has also been revealed by using RAPD markers (*Zhao et al., 2008*) and ISSR markers (*Jiang, Chen & Bao, 2012*). Overall, high outcrossing rate and gene flow facilitate the formation of continuously distributed homogenous populations across its natural range of warm-temperate and subtropical zones, thus resulting in comparatively low population genetic differentiation. With highly polymorphic SSR markers, we robustly identified population genetic structure corresponding to populations' geographical locations. Our results detected three genetic clusters: the northern cluster including the four populations of Yellow River Delta, the fifth Bohai costal population CY, and Beijing population, the second genetic cluster including two populations of Huang Zhou Bay, and the third genetic cluster containing the remaining Hang Zhou Bay population. The northern (warm-temperate

zone) populations were clustered together as one genetic group and populations of Hang Zhou Bay fell into two other genetic clusters, which may imply the different origins of the studied populations of the two different climate zones. Previous studies revealed more nucleus ribosome DNA (nrDNA) haplotypes in the populations of the lower reaches of the Yellow River, implying this species may have dispersed eastward along the direction of the flowing water (*Sun et al., 2019*). The continuously distributed populations in the Bohai Sea coastal flats are possibly the results of the northward and southward dispersal of seeds from Yellow River Delta by sea currents, which has yet to be tested by extensive sampling along coastal flats of the Bohai Sea and new polymorphic markers (*e.g.*, chloroplast microsatellite markers) for population genetic analysis. The inland population FS from Beijing was clustered together with populations of Bohai Bay, which is consistent with a study where a Beijing population is genetically closer to a population of the Bohai Sea coastal flat from Korea (*Lee, Gaskin & Young, 2018b*). The subtropical (the East Sea) populations were genetically different from populations from the Bohai Sea coastal flats, which may be the result of seed dispersal *via* the Yellow River at different times when 150 years ago the Yellow River gorged into the Huanghai Sea instead of the Bohai Sea. Yet we did not include populations from the old estuary flat of the Yellow River in the Huanghai Sea coast in this study.

## Isolation pattern of *T. chinensis*

Using Mantel tests, we detected a highly significant correlations between genetic distance and geographical and environmental variables. Because the autocorrelation between geographical and environmental variables is also high ($R^2 = 0.9357$), we performed partial Mantel tests to disentangle the role of geographic distance and environmental variables in shaping the genetic variation detected by microsatellite markers. Our high correlation ($R^2 = 0.5224$ for IBD and $R^2 = 0.5225$ for IBE) between genetic distance and geographical and environmental variables contrasts with no or low significant correlation ($R^2 = 0.1773$, $p < 0.01$, (*Liang et al., 2018a*); $p = 0.39$, (*Zhu et al., 2016*)) detected before, which may partly be due to their relatively small sampling areas (Yellow River Delta for *Zhu et al., 2016*, and the Yellow River reaches for *Liang et al., 2018a*) and the usage of less variable microsatellite loci.

## CONCLUSIONS

In this study, we evaluated the genetic diversity and population structure of *T. chinensis* using 12 SSR primer pairs. *T. chinensis* was found to maintain a high level of genetic diversity. On average, populations of Yellow River Delta showed a higher level of genetic diversity than those from Hang Zhou Bay. The studied populations showed strong IBD and IBE patterns and significant population genetic structure corresponding to sampling sites was detected despite a quite low level of population genetic differentiation. Our results revealed population genetic structure related to environmental variables of two climate zones in a fast spread shrub species. Further extensive sampling and more informative variable loci analysis will enable us to determine the detailed dispersal routine and better understand genetic variation conferring local environmental adaptations of *T. chinensis*.

### Funding

This work was supported by the Foundation of Linyi Univeristy (No. A5130673). The funders had no role in study design, data collection and analysis, decision to publish, or preparation of the manuscript.

### Grant Disclosures

The following grant information was disclosed by the authors:
Linyi Univeristy: A5130673.

### Competing Interests

The authors declare there are no competing interests.

### Author Contributions

- Zhaoyu Jiang performed the experiments, prepared figures and/or tables, and approved the final draft.
- Aoao Yang performed the experiments, authored or reviewed drafts of the article, and approved the final draft.
- Haiguang Zhang analyzed the data, prepared figures and/or tables, and approved the final draft.
- Wenbo Wang analyzed the data, prepared figures and/or tables, and approved the final draft.
- Ruhua Zhang conceived and designed the experiments, prepared figures and/or tables, authored or reviewed drafts of the article, and approved the final draft.

### Data Availability

Raw data are available in the Supplemental Files.

### Supplemental Information

Supplemental information for this article can be found online at http://dx.doi.org/10.7717/peerj.15882#supplemental-information.

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

# PeerJ

**Zhang Y-Y, Fang Y-M, Mukui K, Li X-X, Xia T. 2013.** Molecular characterization and genetic structure of *Quercus acutissima* germplasm in China using microsatellites. *Molecular Biology Reports* **40**:4083–4090 DOI 10.1007/s11033-013-2486-6.

**Zhao JK, Li-An Xu, Xie HF, Zhao DY, Huang MR. 2008.** Rapd analysis of population genetic diversity of *Tamarix chinensis* in yellow river delta. *Journal of Nanjing Forestry University* **32**(5):56–60.

**Zhu Z, Zhang LY, Gao LX, Tang SQ, Zhao Y, Yang J. 2016.** Local habitat condition rather than geographic distance determines the genetic structure of *Tamarix chinensis* populations in Yellow River Delta, China. *Tree Genetics & Genomes* **12**(1):14 DOI 10.1007/s11295-016-0971-5.

**Zohary M. 1987.** Tamarix L. In: Zohary M, ed. *Flora palaestina 2*. Jerusalem: Israel Academy of Sciences and Humanities, 350–362.