# Peer review of "Population structure and genetic diversity of Tamarix chinensis as revealed with microsatellite markers in two estuarine flats"

_PeerJ, doi:10.7717/peerj.15882_

## Round 0.1 · original submission · Major Revisions

Please revise the article considering the review reports, especially reviewer 2 and 3.

·

Basic reporting

The manuscript lacks the description about importance of Tamarix chinensis. Authors are requested to mention a brief description on ecological or economical importance of the same plant.
This plant species has unique salt tolerance mechanism, authors are requested to incorporate description on the same. Was there any study regarding morpho / physiological characters, if conducted, author may incorporate these parameters.
In manuscript EST SSR markers are used, is there any functional significance of these markers used in the study.

Experimental design

no comments

Validity of the findings

no comments

Reviewer 2 ·

Basic reporting

Study related to deciduous shrub Tamarix chinensis is very important and really interesting. I have gone through the whole manuscript and I’ve found very interesting findings related to variability and diversity of the Tamarix chinensis. I would like to give some common comments regarding this manuscript. It is written well but not organized.

Experimental design

Sampling procedure is very unclear which needs more clarity and detail information with proper reference. Describe why you take 50m distance between two samples with sampling procedure.

Validity of the findings

Result
In the result section many information given regarding various analysis but not a single parameter is explained for detail understanding. Author must give few line or key words to understand importance of parameters. Results is described without its description.
Discussion
Discussion section compare present results with the previous study but not provide the novelty than previous researchers. For example, Fst value is given and compared with previous study but not discussed in present context. Please provide the related information

Additional comments

Introduction
I mentioned that introduction section may required more revision. Initiation of introduction is started from the common population structure. That is not appropriate. Initially in the manuscript, Author must give the identification and introduction of Tamarix chinensis and how it is important for human being and or for environment. How it reached at the endemic stage? How it make difference in the environmental balance. Than give the description of the population found at different regions and discus about its diversity. In this manuscript, introduction section contain some citation regarding population structuring and diversity (Line no 38-48). That is not really important here in this section. For more detail see the annotated manuscript.

Other
Image is not clear
Table 2 required legends for used short forms
Table 3 Formatting for font is required
Figure 2 is not clear
In the supplemental table 1: Provide the full form of the region or locations YHK, CY, FS, YDG, YXX, YHD etc.


This study is good regarding diversity and population genetic analysis. But on my point of view this study used only 8 polymorphic markers are very less. But I hope you used more markers but they were monomorphic. But for this kind of results you may provide the information of monomorphic markers also. Include its information and reanalyzed the data you may get different result. Kindly provide the gel images in a supplementary files and figures.

Annotated reviews are not available for download in order to protect the identity of reviewers who chose to remain anonymous.

Reviewer 3 ·

Basic reporting

Tamarix chinensis is an important tree species and its diversity study is really interesting. Author chooses good aspects for diversity analysis of really important species like this. I have gone through these whole manuscript, and I found some interesting facts regarding diversity analysis. Author used many statistical aspects of molecular diversity analysis in this regards. They used population structure and diversity analysis using many software like GENEPOP, GenAlEx, CERVUS, ARLEQUIN, BOTTLENECK and STRUCTURE etc. I would like to give some suggestions are followed
1. Introduction should more precise and should be focused on the Tamarix chinensis.
2. Authors should clearly mention the objective of their study
3. Only population genetic analysis is not an objective.
4. Author should give more detailed botanical information, uses and importance of Tamarix chinensis for nature.
5. There are many linguistic errors are present. It should be removed before publication

Experimental design

1. Authors did not mention the standard sampling method.
2. Sampling method should be precise and elaborative to understand. Citation is required.
3. Sampling distance is mentioned is very short. I would like to suggest that author should take sample of at least 500m away from each other.
4. Author should provide the different images of different populations and its habitat for more clear idea about observed diversity among populations.
5. They should mentioned their actual habit of growth and different morphological variations of different populations.
6. Provide the annealing temperature of used SSR primers.
7. In line no. 125, they claimed that markers were designed and employed in the study. Then provide the source database and primer related information for each primer in supplementary form.
8. Author used too much various software and extract the results. But due to the large amount of information it is became very difficult to understand the results.
9. They should minimize the parameters for diversity analysis
10. Provide the formula of used parameters for the diversity analysis

Validity of the findings

1. Author must give the brief of the used parameters for understanding
2. They must discuss the extracted result and discuss it with the recent authors.
3. Used tables and figures require more formatting
4. Please provide the full name of parameter in the bottom of all the tables.
5. Provide only the results of your study in brief. They must provide the new and recognized findings with future prospect it is really missing from the manuscript.
6. Explain in few sentences that available results are how important and how use full for the mankind

Additional comments

I would like to suggest to add result of more polymorphic primers if available. If polymorphism is not achieved, I hope you can add information of the other monomorphic markers also.
Some other errors are mentioned here,
Provide the full forms wherever necessary at least one
Line 15: Check spelling of men
Line 125: Rewrite this in understandable format and provide development criteria, database name and other software’s used to design primers.
Line 130: In taq polymerase concentration, unit is incomplete.
Line 126-Line 135: Check the units.
Line 136: Replace PCR profile with PCR conditions.
Line 138: Provide the annealing temperature of all used primers in supplementary table.
Line 144: Explain importance of null allele.
Line 158: Mention unit after 80

---

## Round 0.2 · Minor Revisions

Dear author, there is still scope to improve the article please go through the comments and do needful changes for further consideration of the article.

Reviewer 2 ·

Basic reporting

Basic reporting of the species is written very well and very clearly. There are some modifications needed to increase the fluency in reading and understanding.

Experimental design

It is original and well organized. Lab experiments and software analysis are notable. It is very well modified and reframed in clear version.

Validity of the findings

Previous studies are not mentions the detail information of population structure, which was covered in this study. Some modifications are needed through breaking of long confusing statements.

Additional comments

Study related to deciduous shrub Tamarix chinensis is very important and really interesting. I have gone through the revised manuscript and I’ve found precise writing with more clarity with very interesting findings related to variability and diversity of the Tamarix chinensis. I would like to give some common comments regarding revised manuscript. Revised manuscript is modified in satisfactory way. Even though, there are few long sentence which make confusion. I have annotated the manuscript and few changes suggested.

Annotated reviews are not available for download in order to protect the identity of reviewers who chose to remain anonymous.

Reviewer 3 ·

Basic reporting

Proper but not well arranged

Experimental design

The revised version is good, but some modifications are still needed. refer the annotated manuscript

Validity of the findings

Validity of the findings are presented in confusing manner. Need more improvement

Additional comments

I have gone through the whole manuscript. It was revised as per the requirements but still there are many errors in the text and presentation. Present version is contained many confusing statements which are indicated in the annotated manuscript. In the whole manuscript author used brackets to present the results obtained during study. This is not a proper way to present the data. It makes confusion during reading. There are still many linguistic errors and confusing long statements. These need to be improved before publication. Gel photograph is provided but photograph is partially annotated give the proper labels. More over supplementary tables are incorrectly annotated in the manuscript. I have suggested many improvement in the annotated manuscript please follow it and resubmit.

Annotated reviews are not available for download in order to protect the identity of reviewers who chose to remain anonymous.

---

## Round 0.3 · accepted · Accept

Revised article accepted for publication.

Reviewer 2 ·

Basic reporting

Good and clear

Experimental design

proper

Validity of the findings

impactfull

Additional comments

Line no 25-26: Add the discussion in one line for gene-flow like “Gene flow (Nm) was 4.254
26 estimated from FST indicated the ……”
Line no 28: remove “our results” and add “This study ….
Line no 104: add space in the “insilica”.
Line no 137: add comma between words into and within
Line no 151: Provide the version of STRUCTURE HARVESTER
Line no 203: remove of was and add “were”
Line no 265: Correct it by adding T in manuscript (table 2)
Table no 4: What is Va, Vb and Vc ? Provide valid indication under the table.